# Intrinsic-ControlNet: A Generative Rendering Approach to Render Any Real and Unreal

## Abstract

Rendering highly realistic images from 3D assets is one of the most persistent challenges of the graphics community, which is procedurally conducted by simulating real-world geometry, material, and light transportation. However, such simulations are both burdensome and expensive. Recently, diffusion models have seen great success in realistic image generation by leveraging priors from large datasets of real-world images. Nonetheless, these generative models provide limited control over the output and, unlike graphic pipelines, cannot accurately integrate materials and geometric information for precise image synthesis. In this work, we propose a generative rendering framework, Intrinsic-ControlNet, that enables the generation of corresponding RGB images from 3D assets like a rendering engine by taking intrinsic images, e.g., material, normal, and structural information, as network inputs. We propose a novel multi-conditional control method that allows the model to accept any number of intrinsic images as input conditions. To mitigate bias from synthetic training data, we propose a new model architecture that allows appearance and structural conditions to be input separately into Control-Net Zhang et al. (2023), preserving the realism of appearance generation from real data while maintaining structural control capabilities from synthetic data. Experiments and user studies demonstrate that our method can generate controllable, highly realistic images based on the input intrinsic images.

## 1 Introduction

Despite decades of development, generating photorealistic images in computer graphics remains a highly challenging and expensive task. Specifically, Modeling objects across different scales to create scenes that closely align with the real world is highly challenging, especially given the diverse material properties Cook & Torrance (1982); Ngan et al. (2005) that objects can exhibit. Additionally, simulating light propagation within a scene to solve the rendering equation Kajiya (1986), i.e., the rendering process, requires a significant computational cost. Consequently, existing pipelines struggle with tasks such as rendering highly realistic virtual scenes, inserting virtual objects into real environments, or seamlessly blending and editing real and virtual scenes in a low-cost and efficient way, highlighting the need for a more lightweight solution. Recently, diffusion models (Ho et al., 2020; Dhariwal & Nichol, 2021; Song et al., 2020) have achieved significant success in generating realistic images by utilizing priors from large real-world image datasets. This work inspired us to develop a neural network-based generative rendering framework with two main features. First, it uses neural networks to extract 3D information from input condition images, drawing on prior knowledge from extensive real-world data, thus eliminating the need for explicit 3D modeling and light transportation. Second, it enables flexible adjustment of geometric structure and material properties, similar to a graphics engine. However, current diffusion methods provide only limited control over the generated images and lack the ability to adjust geometric structure and material properties freely.

In this work, we propose Intrinsic-ControlNet, a novel framework that leverages intrinsic scene descriptions commonly used by rendering engines, including properties like albedo, normal, depth, metallicity, and roughness, among others, to generate controllable output images. Our method presents a novel generative rendering approach that combines data-driven techniques with traditional rendering controls, allowing us to create highly realistic images without exhaustive physical

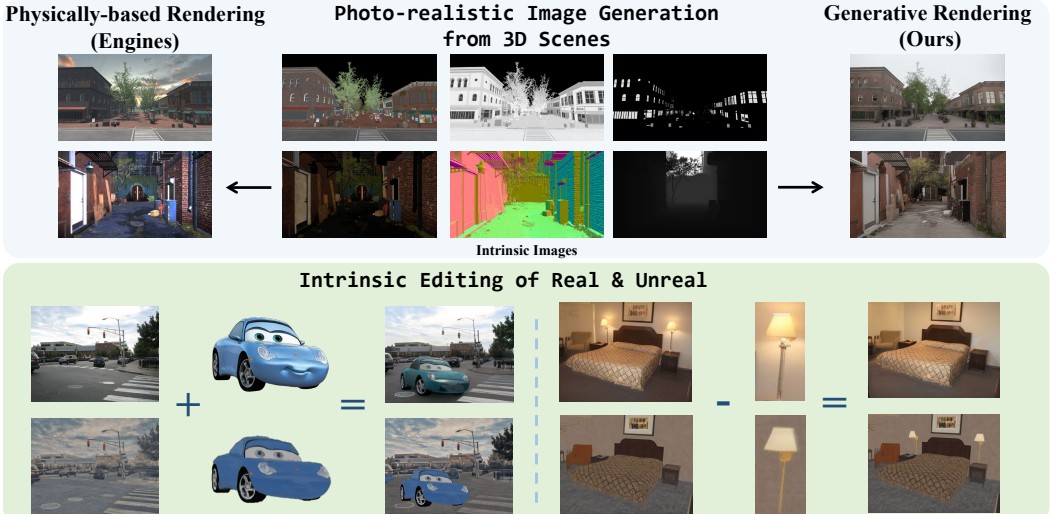

Figure 1: We propose a generative rendering framework that synthesizes realistic images from 3D assets, similar to a rendering engine, using intrinsic images like material, normal, and structural information as network inputs. This approach greatly simplifies intrinsic editing tasks for real images, such as object insertion and removal, while achieving better results compared to existing methods.

modeling or computationally intensive rendering. Instead of relying on physics-based analytical calculations, our approach utilizes a diffusion model to generate photorealistic images.

To realize our generative rendering framework, we need to address the following challenges. First, it is challenging to obtain all the pairwise intrinsic images needed for training, and the same issue arises during inference. This requires our model to support any combination of intrinsic images as conditional control. To address this, we designed a switch structure that concatenates an uncertain number of condition images along the channel, enabling the model to generate convincing images under different input combinations. Additionally, the use of multiple modalities of condition image inputs requires effective feature extraction to prevent diminished loss for individual modalities. To address this, we introduced residual blocks into the encoder, allowing more detailed features to be captured during the ControlNet Zhang et al. (2023) training process, thereby producing higher-quality images.

Secondly, our goal is to generate highly realistic target images, but the existing pairwise intrinsic images are synthetic. There remains a domain gap between the target images in the data and real-world images. To address this, incorporating real-world data into the training process is essential for enabling the model to generate realistic images. However, unlike synthetic data, there is a vast amount of real-world images that lack corresponding pairwise intrinsic images. To address this, we utilized a pre-trained diffusion model to predict the corresponding intrinsic images from real-world RGB images. Obviously, the predicted intrinsic images are not entirely accurate. However, fortunately, the advantages of synthetic data complement the limitations of real data, as synthetic data provides accurate intrinsic images while real-world data offers highly realistic target images. Based on this observation, we split the input condition images into two parts: appearance and structure, with each being processed by separate ControlNets to provide appearance and structure control for the latent diffusion model. During training, the appearance part is sourced entirely from the real image dataset, while the structure part comes from both real and synthetic datasets. Through this design, we eliminate the appearance bias in synthetic data and the inaccuracies in structural information from real data, enabling the generated images to achieve high realism while maintaining controllable structures. Additionally, we observe that the convergence speeds of appearance and structure in the diffusion model differ. Employing two distinct ControlNets to manage the conditional features separately helps the model achieve stable convergence for both appearance and structure.

Ablation experiments confirm that our network design and training strategy significantly enhances the generation of highly realistic images. Furthermore, experiments and user studies demonstrate

that our framework can produce controllable, editable, and highly realistic images, even surpassing graphics engines in terms of realism. In summary, our contributions are as follows:

- We propose a generative rendering framework, Intrinsic-ControlNet, which allows for the controllable and editable generation of highly realistic RGB images from 3D assets, similar to a rendering engine, by using intrinsic images as network inputs.

- We propose a novel multi-conditional control method that allows the model to accept any number of intrinsic images as input conditions.

- We introduce a new model architecture that separates appearance and structural conditions in ControlNet, enabling the preservation of realistic appearance from real data while retaining precise structural control from synthetic data.

## 2 RELATED WORK

**Separating appearance and structure.** In tasks involving multi-condition control, many existing methods separate the conditions that control appearance and structure to guide the generation of results, significantly improving the flexibility and quality of control. For example, Ye et al. (2023) combines ControlNet (Zhang et al., 2023) and T2I-Adapter (Mou et al., 2024) to separately control structure and appearance style. Mo et al. (2024) uses structural guidance in the subspace to enforce alignment with input conditions, while applying appearance guidance at the same image level for appearance transfer. Lin et al. (2024a) achieves appearance transfer and strict structural control during inference through feature injection and self-attention correspondence. Although these methods can achieve appearance transfer while maintaining structural features, their control conditions are relatively singular, i.e., a single style image and a single structure image as input conditions. Our method, however, introduces a multi-way switch and two separate appearance and structure branches, enabling the generation of images guided by up to six intrinsic images.

**Neural image editing.** Advancements in generative models, particularly Generative Adversarial Networks (GANs) (Goodfellow et al., 2014; Karras et al., 2019; 2020) and diffusion models (Ho et al., 2020; Nichol & Dhariwal, 2021; Rombach et al., 2022), have substantially enhanced various image editing capabilities, including style transfer , image-to-image translation, and manipulation within latent spaces (Huang & Belongie, 2017; Phillip et al., 2017; Schuhmann et al., 2022). The introduction of text-guided diffusion frameworks (Nichol et al., 2021; Saharia et al., 2022) has further expanded the scope of image modifications by allowing users to influence edits through natural language prompts, thereby improving both control and user experience. In the area of object insertion, more recent works have harnessed deep generative models to predict realistic object locations, as seen in Compositing GAN (Azadi et al., 2020) and OBJect3DIT (Michel et al., 2024), which address the complexities of integrating objects into diverse and intricate real-world scenes. However, existing methods often perform editing operations in the pixel space. In contrast, our method takes multiple intrinsic images as input and leverages the latent diffusion model to extract underlying 3D scene, generating images at the scene level rather than merely performing pixel-level editing.

**Conditional image generation.** GANs (Creswell et al., 2017) are generative models capable of conditional control, but their training processes are relatively unstable and require significant amounts of training time and resources. The emergence of diffusion models (Ho et al., 2020; Dhariwal & Nichol, 2021; Song et al., 2020) has reduced training complexity and improved generation quality. Early studies (Ho et al., 2020) primarily focused on unconditional generation tasks, where the models generated images solely based on the initial noise. However, with the growing demand for more precise generation tasks, researchers started investigating how to incorporate additional information during the generation process to achieve more accurate control over the results. The initial text-to-image diffusion models (Ramesh et al., 2022; Gu et al., 2022; Podell et al., 2023; Ding et al., 2021; Zhou et al., 2022; Ramesh et al., 2021b; Mou et al., 2024) attracted significant attention by allowing control over the appearance of generated images through text prompts. Furthermore, some models (Zhang et al., 2023; Ye et al., 2023; Zhao et al., 2024; Wang et al., 2024) have extended the control conditions from purely text prompts to include one or more images, thereby further enhancing control over the generated results and expanding the applications range of the model. Zeng et al. (2024) encodes multiple conditions into the latent space and then inputs them into a latent diffusion model to control image generation. However, existing methods still face some challenges when us-

ing multiple control conditions as inputs. Our method successfully handles multiple intrinsic images as input conditions to generate controllable, photorealistic images.

# 3 METHODOLOGY

We first describe the target problem addressed in this paper in Section 3.1, followed by a detailed explanation of how our framework solves this problem in the subsequent sections. The core of our method is shown in Figure 2. Intrinsic-ControlNet accepts any combination of intrinsic images as input, including albedo, metallicity, roughness, normal, depth, and semantic segmentation. They are integrated via a multi-way structure (Section 3.2) to form a comprehensive control condition, then extract features through a tailored encoder. Notably, the input condition conditions are split into two groups, i.e., appearance and structure, and are then sent separately to the respective ControlNets responsible for managing appearance and structure control. Various factors, such as biases in the training data and differences in the convergence speeds of different conditions, led us to adopt this design (Section 3.3). Furthermore, we will introduce the training procedure for Intrinsic-ControlNet (Section 3.4).

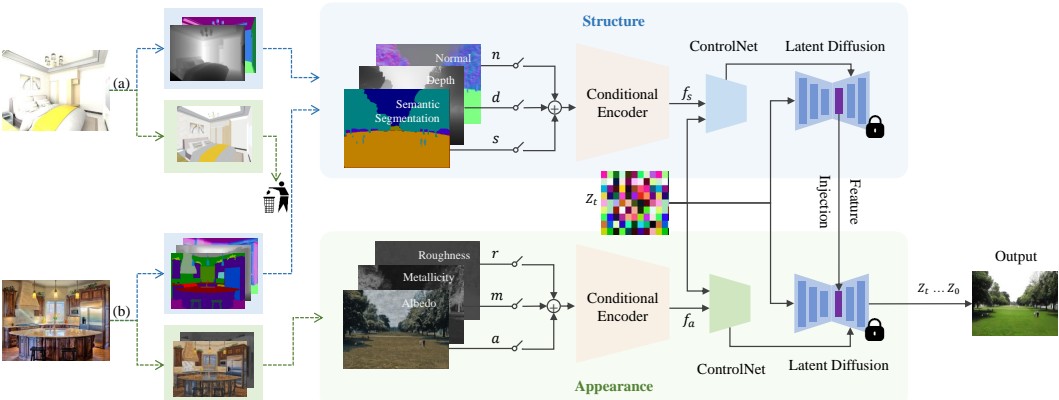

Figure 2: Overview of our pipeline. Given the intrinsic images from the graphic engine (a) and the other predicted from Kocsis et al. (2023) (b), they are divided into two groups, appearance, and structure, and processed through a multi-way switch to form the comprehensive control conditions (Section 3.2). Then, a tailored conditional encoder is employed to extract the diverse condition features. The features are fed separately into two ControlNets Zhang et al. (2023), which individually control the latent diffusion model to generate the appearance and structure information in the final realistic images (Section 3.3).

## 3.1 PROBLEM STATEMENT

Our goal is to achieve a novel generative rendering framework that leverages intrinsic scene descriptions commonly used by rendering engines, including properties like albedo, normal, depth, metallicity, and roughness, among others, to generate controllable and highly realistic images. Traditional graphics engines have long been dedicated to solving similar problems:

$$I = R(a, r, m, n, d, o, l), \tag{1}$$

where $R$ represents the rendering layer, and $a, r, m, n, d$ correspond to albedo, roughness, metallic, normal, and depth within the current viewport. $o$ represents the structural information of the entire scene outside the viewport, and $l$ corresponds to the lighting in the scene. However, traditional engines rely entirely on manually simulating the physical world, including geometry, materials, and lighting, without incorporating real-world data into the final image generation. As a result, it remains an in-domain generation problem where realism is within the limit of input 3D models. In fact, the goal of the synthesis task is to generate photo-realistic images, which involves tackling a cross-domain generation problem as shown in Figure 3. The input condition intrinsic images are derived either from synthetic data produced by modeling engines in the unreal domain while the

target is a photorealistic image in the real domain. In our framework, knowledge from the real world is introduced into the image generation process through the latent diffusion model:

$$I = D(f_a, f_s, f_t \mid \epsilon_\theta(\cdot)), f_a = E(a, r, m), f_s = E(n, d, s), f_t = C(t), \qquad (2)$$

where $f_a$ and $f_s$ represent the appearance information and structure information, respectively, extracted from the intrinsic images using the condition encoder $E$. $s$ corresponds to the segmentation map, while $t$ represents the text prompt. In addition, $f_t$ represents the global structural information and $C$ denotes the text encoder of the latent diffusion model, which aligns with the information contained in $o, l$ in Equation (2). Specifically, $D$ represents our framework, and $\epsilon_\theta(\cdot)$ refers to the latent diffusion model, or more precisely, the prior knowledge it contains from the real world. In the upcoming sections, we will explain in detail how to resolve the problem outlined in Equation (2).

## 3.2 IMAGE ENCODING AND MULTI-CONTROL WITH MULTI-WAY SWITCH

**Condition image encoder.** Extracting effective condition features from various modalities of intrinsic images is crucial for controlling image generation. The image encoder of the original ControlNet Zhang et al. (2023) consists of eight convolution layers. While this simple design excels at extracting high-level, holistic features from images, it struggles to capture the detailed features in condition images, hindering our framework from producing realistic images that can rival those generated by traditional graphics pipelines. Inspired by Luo et al. (2024), We replace the convolution layers in the original image encoder with residual blocks $E(\cdot)$ (He et al., 2016). Therefore, the output of the image encoder, $c_I^* = E(I)$, serves as the

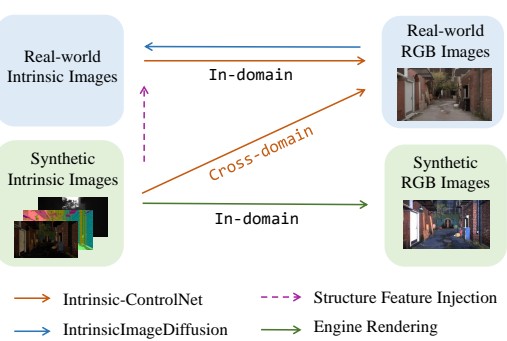

Figure 3: Our cross-domain rendering.

condition for ControlNet. The residual blocks gradually extract abstract, high-level features from the input condition image while preserving and integrating fine details into the final features. This ensures that the condition features can precisely control the detailed aspects of the generated image.

**Multi-way condition switch.** Equipped with the features extracted from multiple condition images, we need to design a structure that utilizes them to guide the latent diffusion model in generating images that align with the specified conditions. One straightforward approach is to train a separate ControlNet for each condition image, then combine the control features from all the ControlNets before injecting them into the latent diffusion model. We refer to this approach as Multi-ControlNet in the following sections. However, this approach requires manually assigning blending weights for each condition when mixing the condition features, depending on the influence of each condition on the final generated image, which is a highly complex task. Secondly, training a ControlNet for each intrinsic image significantly increases training time and the network size, especially when working with six intrinsic images in our task. To address this, we designed a multi-way switch that enables the use of any combination of intrinsic images as conditions during both training and inference. Specifically, each intrinsic image has two possible states, when selected, the original image is used in subsequent processing, and when not selected, it is replaced with a black image of the same size as a placeholder. The intrinsic images are then concatenated to form a multi-channel condition input $c_I^{multi-way*}$. Due to the presence of placeholders, the number of channels in $c_I^{multi-way*}$ remains fixed. This multi-channel condition input is then fed into the condition encoder, generating a fully integrated condition feature, which is subsequently passed to the corresponding ControlNet to produce the control features for the latent diffusion model. With the multi-way switch, multiple intrinsic image conditions can be controlled by a single ControlNet, avoiding the need for multiple ControlNets. Additionally, condition weights are automatically optimized using pairwise training data, removing the bias from manual weight mixing.

## 3.3 APPEARANCE AND STRUCTURE CONTROL

To address the cross-domain rendering, we split the control paths of Intrinsic-ControlNet into two branches, appearance and structure (referred to as A&S for brevity), and introduced a structure feature injection design to tackle the problem. The following sections will explain the rationale and detail behind these designs.

**Cross-domain training data.** As shown in Figure 3, our task is to generate photo-realistic images from synthetic intrinsic images, but it is clear that we cannot obtain such pairwise data. The existing pairwise intrinsic images are synthetic data, and if we train the network using them directly, the generated images will exhibit a significant domain gap from real-world images as shown in Figure 10. Incorporating real-world data into the training process is crucial to ensure the model can produce realistic images. However, unlike synthetic data, while there is a vast amount of real-world images, they lack corresponding pairwise intrinsic images. To address this, we utilized a pre-trained diffusion model Kocsis et al. (2023) to predict the corresponding intrinsic images from real-world RGB images. Similarly, since it is impossible to obtain real intrinsic images paired with real target images for training, the intrinsic images predicted by the diffusion model also contain biases. Training the model directly with such data would lead to generated images with many structural inaccuracies as shown in Figure 10. Fortunately, the advantages of synthetic data complement the limitations of real data, as synthetic data provides accurate intrinsic images while real-world data offers highly realistic target images. Based on this observation, we divide the input condition images into two parts, i.e., appearance and structure, and feed them into separate control branches. During training, the appearance part is sourced entirely from the real image dataset, while the structure part comes from both real and synthetic datasets. The final image is generated directly by the appearance branch, with the structure branch sharing its features via structure feature injection without directly contributing to the final output. Through this design, we eliminate the appearance bias in synthetic data and the inaccuracies in structural information from real data, enabling the generated images to achieve high realism while maintaining controllable structures.

**Structure feature injection.** As mentioned earlier, to avoid introducing appearance bias from synthetic data into the final generated image, we prevent the structure branch from directly contributing to the image generation. However, we still aim to maintain precise structural control over the images produced by Intrinsic-ControlNet. Previous work Tumanyan et al. (2023); Kim et al. (2023); Mo et al. (2024) has observed that diffusion features contain rich layout information. By performing feature and self-attention injection during the diffusion denoising steps, it is possible to control the appearance and structure of generated images without additional training. Inspired by Lin et al. (2024a), we apply structure feature injection during both the training and inference processes. At each time step $t$, we replace the appearance feature $f_a^l$ with the corresponding structure feature $f_s^l$ at the $l$ layer of the latent diffusion model. The features $f_a^l$ and $f_s^l$ are derived from the appearance and structure branches, respectively. This approach ensures that the structure in the image generated by the appearance branch aligns with the structure predicted by the structure branch.

**Independent convergence of A&S.** ControlNet Zhang et al. (2023) introduced the sudden convergence phenomenon observed during training, where the model does not gradually learn the control conditions but instead abruptly succeeds in following the input conditioning image, followed by slow convergence thereafter. Moreover, we observe that the convergence speeds of appearance and structure in the diffusion model differ. Structural control information is generally easier to learn and tends to converge earlier than appearance control. This further motivates us to feed the appearance and structure control information from the intrinsic images into two ControlNets with independent network weights. This approach ensures that each ControlNet converges independently to its optimal point. The results of the convergence comparison between our method and ControlNet can be found in Figure 13.

## 3.4 TRAINING STRATEGY

The typical diffusion process of the latent diffusion model Rombach et al. (2022) and Control-Net Zhang et al. (2023) is mathematically represented as follows:

$$z_t = \sqrt{\overline{\alpha_t}}z_0 + \sqrt{1 - \overline{\alpha_t}}\epsilon, \epsilon \sim \mathcal{N}(0, I), \tag{3}$$

where $z_t$ is the noisy latent feature at time step $t$, $z_0$ is the initial data in latent space, $\epsilon$ is the Gaussian noise, and $\overline{\alpha_t}$ is the parameter of noise strength at time step $t$. In our framework, we employ the V-

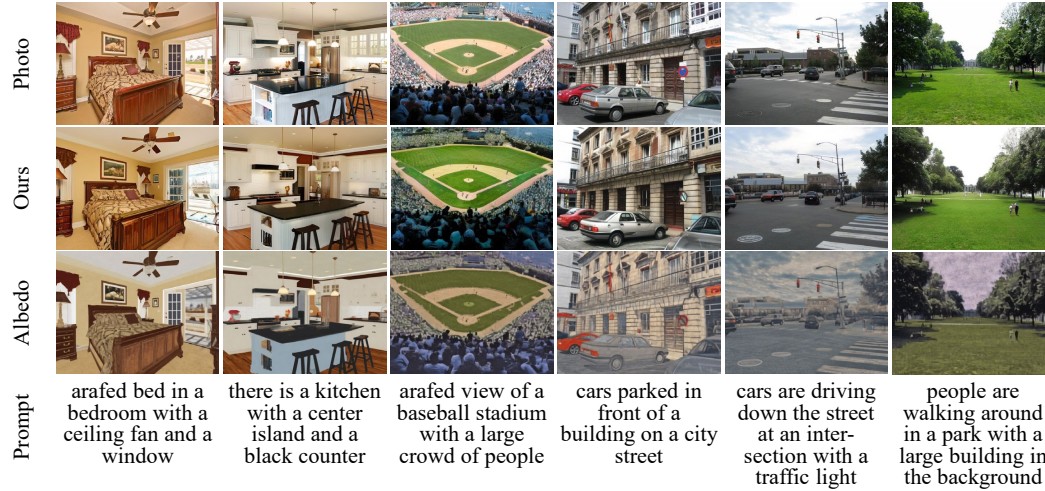

| | | | | | |
|---|---|---|---|---|---|
| arafed bed in a bedroom with a ceiling fan and a window | there is a kitchen with a center island and a black counter | arafed view of a baseball stadium with a large crowd of people | cars parked in front of a building on a city street | cars are driving down the street at an inter­section with a traffic light | people are walking around in a park with a large building in the background |

Figure 4: Our framework uses predicted multiple intrinsic images to generate realistic images that resemble the original. We put other intrinsic images, like normals, in the appendix to save space.

prediction strategy and enforce Zero Terminal SNR Lin et al. (2024b) during the sampling processes to make sure the training behavior is aligned with inference:

$$v_t = \sqrt{\overline{\alpha}_t}\epsilon - \sqrt{1 - \overline{\alpha}_t}z_0, \epsilon \sim \mathcal{N}(0, I). \tag{4}$$

During this denoising process, our framework learn to predict the noise at time step $t$, with the MSE loss:

$$\mathcal{L} = \mathbb{E}_{z_0,y,\epsilon,t,c_I^{multi-way*}}\left[\left\|v_t - \tilde{v}_\theta(z_t, t, \tau_\theta(y), c_I^{multi-way*})\right\|_2^2\right], \tag{5}$$

where $c_I^{multi-way*}$ is the selected condition embeddings after the multi-way switch, and velocity $v_t$ is predicted in diffusion model at time step $t$ instead of the predicted noise. A text prompt $y$ is converted into a sequence of vectors using a text encoder $\tau_\theta(\cdot)$ and mixed with the attention layers of the U-Net $\epsilon_\theta(\cdot)$.

## 4 EXPERIMENTS

### 4.1 DATASETS

Our model is trained on a mix of datasets: (1) 5K real-world indoor ("home or hotel") images and 5K outdoor ("nature landscape" and "urban") images from ADE dataset (Zhou et al., 2017); (2) 4K synthesized indoor images from InteriorVerse (Zhu et al., 2022); (3) 3K synthesized outdoor images from GTA-V dataset (Richter et al., 2016). For real-world data, we generated its intrinsic images (albedo, metallicity, roughness, normal, and depth) by using the pre-trained diffusion models pro­vided by Kocsis et al. (2023) and generated its semantic segmentation by using the model provided by Wang et al. (2022). For each image, we resize it to the resolution of $512 \times 512$ pixels and use BLIP model (Li et al., 2022) to generate the corresponding text prompt.

Table 1: Quantitative comparison of different methods and different intrinsic images.

| Intrinsic | All | | A. | | M.+R. | | A.+N.+D.+S. | |
|---|---|---|---|---|---|---|---|---|
| Scores | CLIP ↑ | L2 ↓ | CLIP ↑ | L2 ↓ | CLIP ↑ | L2 ↓ | CLIP ↑ | L2 ↓ |
| GT | 0.2733 | - | - | - | - | - | - | - |
| Ours | **0.3137** | **128.1** | **0.3011** | **136.7** | **0.3048** | **148.6** | **0.3088** | **133.3** |
| Multi. | 0.2891 | 151.5 | 0.2984 | 185.0 | 0.2960 | 180.5 | 0.3037 | 149.2 |

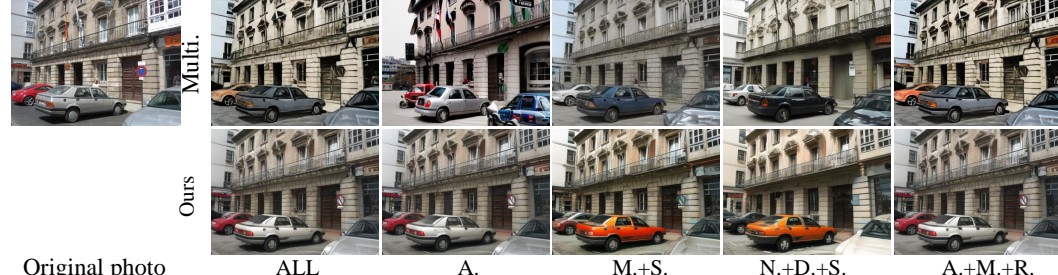

Original photo     ALL     A.     M.+S.     N.+D.+S.     A.+M.+R.

Figure 5: Comparison results of using various combinations of intrinsic conditions as input on real data. 'Multi.' refers to Multi-ControlNet, while A., M., R., N., D., and S. represent albedo, metallic, roughness, normal, depth, and segmentation, respectively.

Table 2: Quantitative comparison of different methods on synthetic datasets.

| Scene | Buildings | | Corner | | Street | | Kitchen | | Park | | Alley | |
|---|---|---|---|---|---|---|---|---|---|---|---|---|
| Scores | CLIP ↑ | User ↑ | CLIP ↑ | User ↑ | CLIP ↑ | User ↑ | CLIP ↑ | User ↑ | CLIP ↑ | User ↑ | CLIP ↑ | User ↑ |
| Ours | **0.266** | **96.4** | **0.258** | **100** | **0.261** | **60.7** | **0.270** | **75.0** | **0.294** | **82.1** | **0.307** | **89.3** |
| Multi. | 0.251 | 0.00 | 0.239 | 0.00 | 0.253 | 14.3 | 0.269 | 3.57 | 0.282 | 14.3 | 0.247 | 0.00 |
| UE | 0.252 | 3.57 | 0.230 | 0.00 | 0.251 | 25.0 | - | - | - | - | - | - |
| Falcor | - | - | - | - | - | - | 0.239 | 21.4 | 0.286 | 3.57 | - | - |
| Blender | - | - | - | - | - | - | - | - | - | - | 0.245 | 10.7 |

## 4.2 EVALUATION RESULTS

**Evaluate on synthetic datasets.** For synthetic data, we compared our method to several popular graphics engines, including UE (Epic Games), Falcor (Benty et al.), and Blender (Blender Foundation), to evaluate image synthesis results. For our method, we employ the GT intrinsic images as input conditions to generate the target image, while for rendering engines, we utilize complete 3D scenes to render the target images. As shown in 6, various methods produced highly realistic photorealistic images. However, compared to the images generated by various engines, our method produces images that more closely align with the realism of human cognition. To better quantify the realism of the generated images, we additionally computed the CLIP (Contrastive Language-Image Pre-training) score Radford et al. (2021); Ramesh et al. (2021a) and conducted a user study for generated results(The detail can be found in Section A.3). We present the quantitative analysis results in Table 2, showing that our method outperforms others in both CLIP scores and user studies.

**Evaluate on real-world datasets.** We evaluated Intrinsic-ControlNet with untrained real-world data from the ADE dataset, including indoor and outdoor data. Specifically, we first predicted intrinsic images from real-world images and used them as inputs to generate new images, then compared the generated images with the original ones to evaluate their similarity. As shown in figure 4, even though the input intrinsic images are biased due to prediction, Intrinsic-ControlNet can still generate photorealistic images that closely resemble the originals. In addition, we compare our method with the Multi-ControlNet approach. To ensure a fair comparison, we train a separate ControlNet Zhang et al. (2023) for each intrinsic image type, namely albedo, normal, depth, metallicity, roughness, and semantic segmentation, on our entire mixed dataset. Then, We separately use our model and the Multi-ControlNet approach to generate images with various combinations of condition inputs. We report the quantitative results of different methods in Table 1. The $L2$ is the metrics introduced by Meng et al. (2021) to quantify faithfulness, which calculates the $L2$ distance summed over all pixels between the guide and the edited output image normalized to [0,1]. As shown in Figure 5 and Table 1, our model performs better than Multi-ControlNet in both single condition control and multiple condition combinations.

Table 3: Quantitative ablation study. Discussion and visualization are in the Appendix.

| | Ours | Only synthetic | Only real | w/o seperate S.& A. | w/o encoder | w/o 0-SNR |
|---|---|---|---|---|---|---|
| CLIP ↑ | **0.3137** | 0.2933 | 0.2953 | 0.2975 | 0.2954 | 0.2997 |

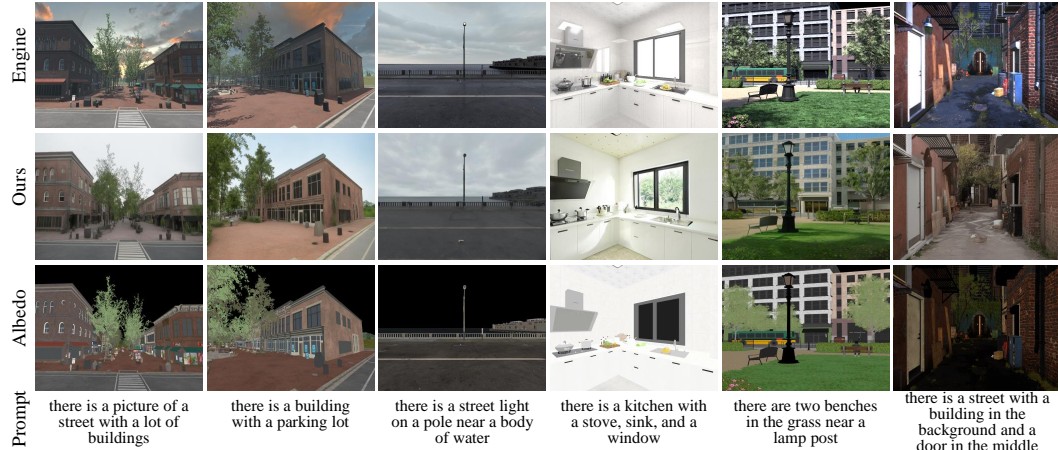

Figure 6: Comparing results of our framework and graphics engines on synthetic datasets.

**Ablation study.** We conducted ablation experiments on our framework design to validate its effectiveness. The quantitative results are shown in Table 3, and more visual qualitative analysis and discussion can be found in the appendix Section A.2.

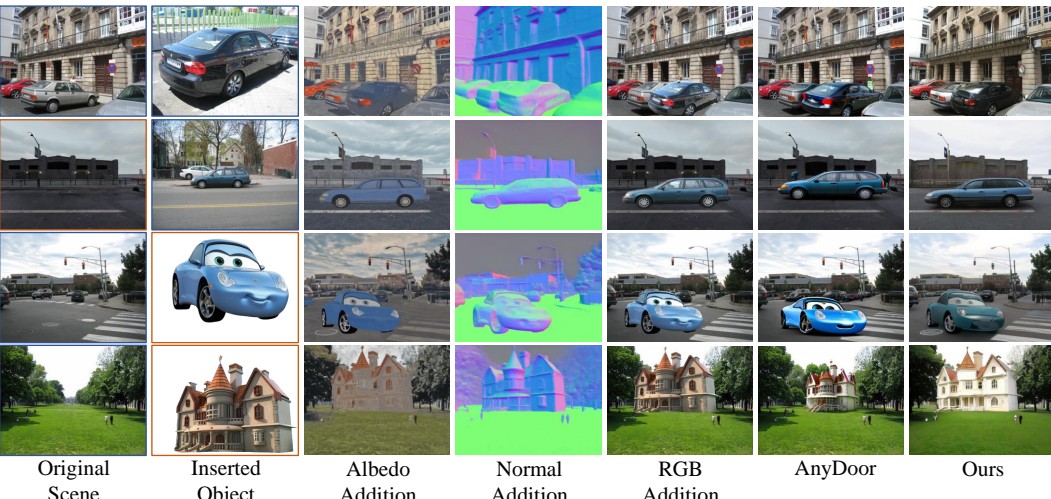

Figure 7: Object insertion, the blue boxes in the first column represent real-world images, while the red boxes represent synthesized images.

### 4.3 APPLICATION

Once our model is fully trained, thanks to comprehensive control conditions, we can perform various editing tasks by modifying intrinsic images, such as changing the material and color of objects or inserting new objects into the scene, while generating highly realistic images. Notably, the entire process does not require 3D scene reconstruction, as it only involves simple modifications to the intrinsic images. The intrinsic images for real-world scenes involved here are all generated by approach Kocsis et al. (2023).

**Object insertion.** In Figure 7, we show the results of various forms of object insertion tasks using our framework, including inserting new real objects into real-world scenes, inserting real objects into synthetic scenes, and inserting virtual objects into real-world scenes. We concatenate the albedo and normal images of two different scenes or objects and then feed them into our framework as condition inputs, generating the corresponding realistic image. We compare our method with the AnyDoor Chen et al. (2024). As shown in Figure 7 and Table 4, our method achieves better results in both qualitative and quantitative evaluations. The inserted objects in our generated images display lighting effects that are consistent with the original scene, such as highlights and shadows, rather than appearing out of place as would happen with simple copying in RGB images.

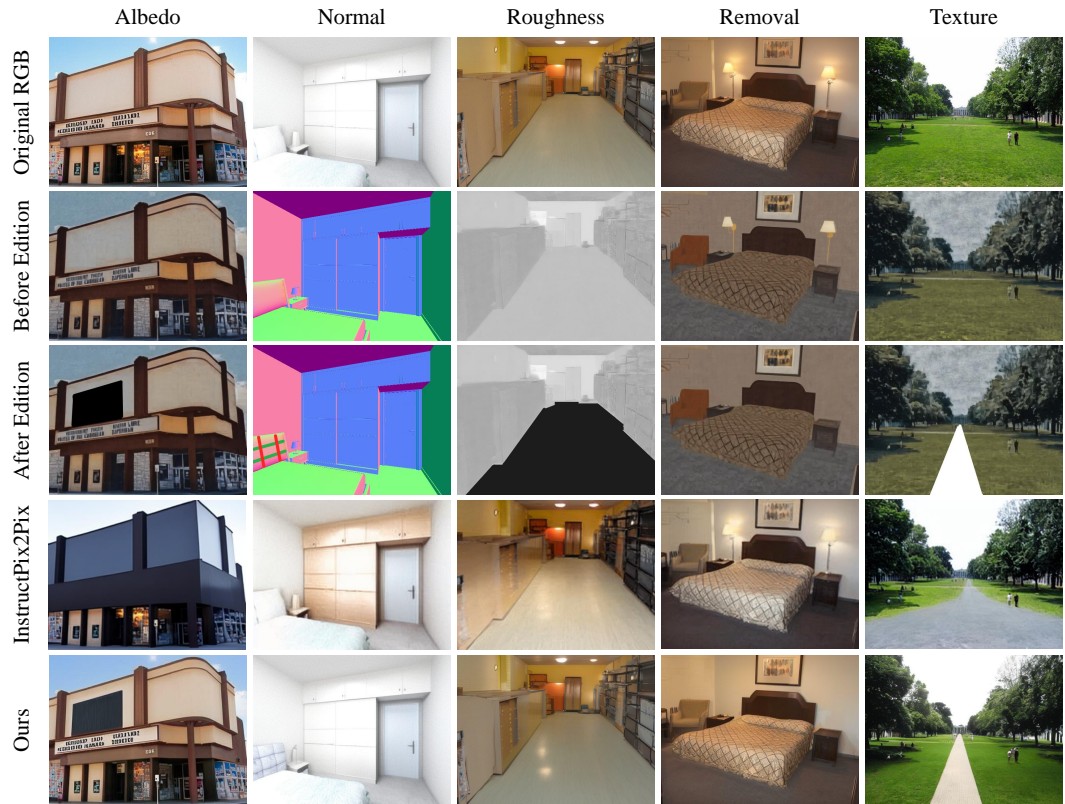

Figure 8: Edition results of real-world and synthetic scenes.

**Scene editing.** In Figure 8, we present the results of scene editing using our method. We can modify the color or texture in the albedo, adjust the structure in the normal, or change roughness values. For example, in the third column of Figure 8, changing the roughness of the floor from rough to smooth creates highlights when light hits it. In the fourth column, we removed the lamp from the albedo image, and we were surprised to find that in the generated RGB image, not only was the lamp itself removed, but its lighting effects on the environment were also eliminated. This demonstrates that our network can implicitly extract information about the entire 3D scene from the intrinsic images, rather than simply performing pixel-level stitching. We compare our method with InstructPix2Pix Brooks et al. (2023), a text-guided editing diffusion model. As shown in Figure 8 and Table 4, our method achieves better results in both qualitative and quantitative evaluations.

Table 4: Quantitative comparison of object insertion and scene editing tasks in different methods.

| Application | Object insertion | | | Scene editing | |
| --- | --- | --- | --- | --- | --- |
| Methods | RGB addition | Ours | AnyDoor | Ours | InstructPix2Pix |
| CLIP ↑ | 0.282 | **0.296** | 0.289 | **0.299** | 0.282 |

## 5  CONCLUSION

In this paper, we present Intrinsic-ControlNet, a novel generative rendering framework that accepts synthetic intrinsic images as input and generates photo-realistic images. To tackle this cross-domain rendering task, we introduce an encoder with a residual structure and a multi-way switch to effectively extract and integrate control features from the condition images. To address the cross-domain issue in the data, we separate appearance and structure, using two distinct ControlNet branches to control the generation. Additionally, a feature injection mechanism ensures that the appearance branch remains unaffected by the appearance features from the structure branch while still producing images with controllable structural details. Experiments demonstrate that our method can render high-quality photorealistic images from intrinsic images while also being effective for various downstream applications such as scene material editing, object insertion, and removal.

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

## A  APPENDIX

### A.1  LIMITATION AND FAILED CASE

Our model generates RGB images with the desired features from intrinsic images in most cases, but it does show some limitations and failure cases. For real-world images, where accurate intrinsic images are unavailable, we generate them using the pretrained model IntrinsicImageDiffusion (Kocsis et al., 2023). If the normal or depth maps generated by IntrinsicImageDiffusion lose structural information, as shown in the first two rows of Figure 9, our results may have similar overall colors to the ground truth but show disorganized local structures. Fortunately, this component can be updated and replaced with a more advanced model to produce more accurate intrinsic images. Additionally, the last two rows of Figure 9 show that when adjacent colors in the albedo map are very similar, our model has difficulty distinguishing between them, causing the generated output to be rendered as a single color.

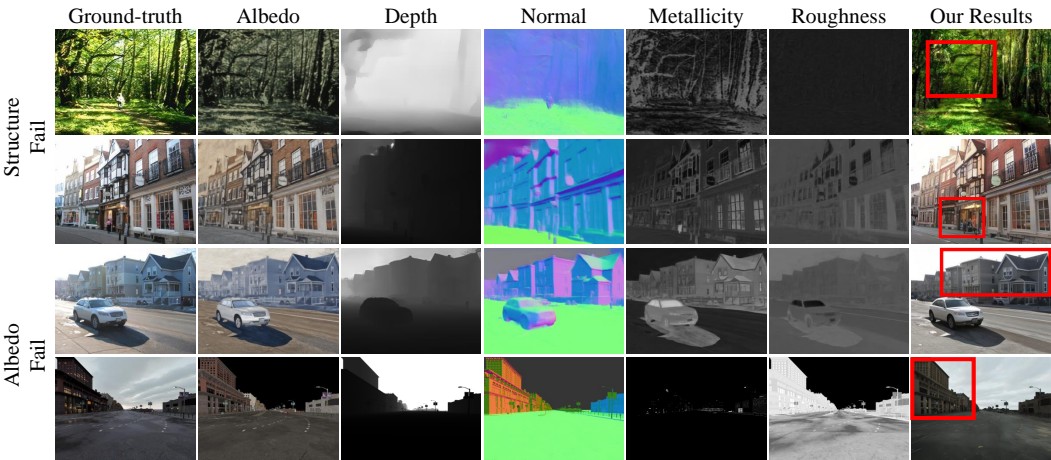

Figure 9: Some failure cases of our framework.

### A.2  ABLATIVE STUDY

**Effect of cross-domain training data.** To evaluate the impact of cross-domain training data on the generated results, we trained our framework separately on synthetic data and real data and show the results in Figure 10. Note that all intrinsic images were used in the training process without excluding the appearance data from the synthetic dataset, and the network structure remained consistent with the full model. As shown in the third column of Figure 10, the model trained without real data tends to produce overall darker images. While the structure is maintained, the results look highly unrealistic, with harsh highlights and a noticeable loss of realism compared to our method. In contrast, when training solely on real data, the model generates more realistic images, but many structural details in the scene are either incorrect or blurred. For example, in the fourth column of the first row in Figure 10, the chandelier frame is distorted, and various objects on the stove appear blurred. Only our approach enables the generated images to achieve high realism while preserving precise, controllable structures.

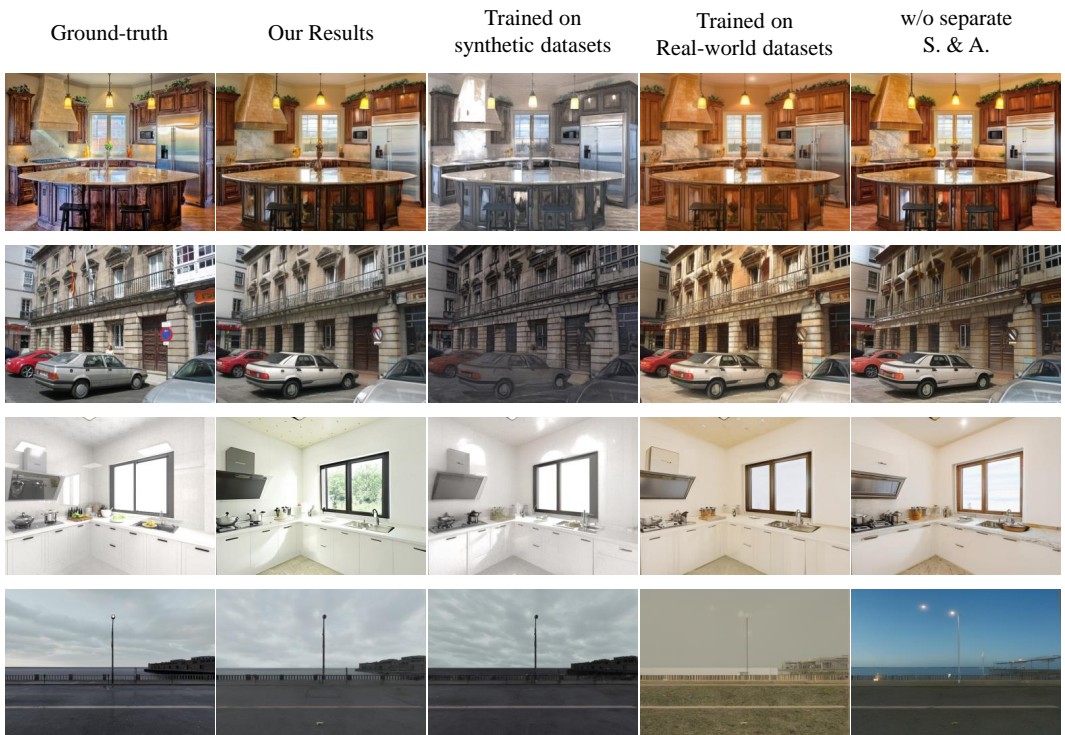

Figure 10: Ablation results of our framework design, with further discussion available in Section A.2.

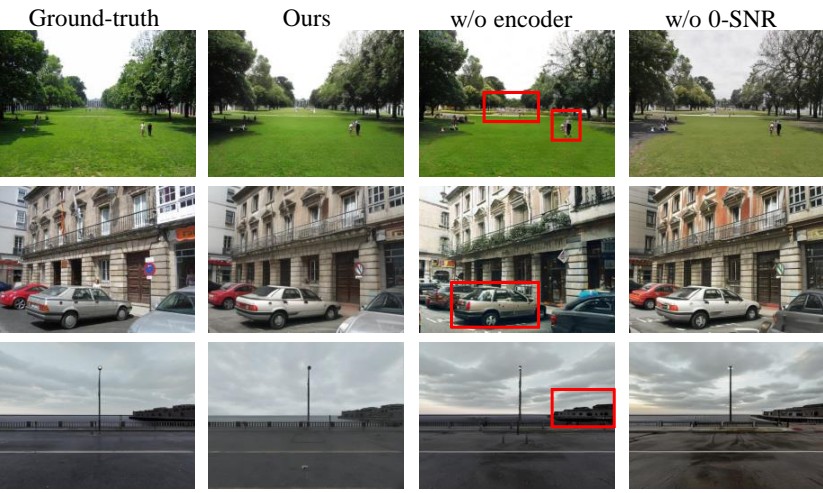

Figure 11: Ablation study on image encoder and 0-SNR strategy.

**Effect of the separate design of appearance and structure.** As mentioned in Section 3.3, to address the cross-domain training challenge, we divided the input intrinsic images into two groups and used two separate branches to manage each group. To validate the effectiveness of this design, we train a model on the same dataset as ours but with a single branch. In this model, all intrinsic images pass through the same ControlNet to control the latent diffusion model. As shown in the fifth column of Figure 10, removing our separation design leads to errors in both color and structure in the generated images, such as the color of the sky in the fourth row and the second-floor windows of the building in the second row.

**Zero terminal SNR strategy.** As shown in Figure 11. The results without using zero SNR strategy in the last column have a significant color discrepancy compared to the ground truth in the first column, appearing overall darker and redder, whereas the results using the zero terminal SNR strategy (Ours) show almost no color deviation. The zero terminal SNR strategy ensures consistency in the diffusion process between training and inference by fully adding noise during training. This closely matches the initial step of the inference process, resulting in generated images that align with the original data distribution.

**Image encoder.** We compare the inference results obtained with and without using the condition image encoder mentioned in Section 3.2 and show the results in Figure 11. With the original image encoder from ControlNet (Zhang et al., 2023), there are clear structural and appearance inaccuracies compared to the ground truth, as shown in the third column. However, our results in the second column preserve the local structures present in the ground truth. The experiment shows that our improved image encoder has a clear advantage in extracting fine structures from intrinsic images, allowing our network to learn more precise control over the details in the synthesized images.

### A.3 METRICS AND USER STUDY

**CLIP score.** The CLIP score is determined by calculating the cosine similarity between the image embedding and the text embedding, which is particularly useful for tasks requiring cross-modal comprehension. For our task, we use the BLIP model (Li et al., 2022) to extract the text prompt from the image synthesized by the graphics engine and then incorporate the prompt with the keywords 'photo-realism' to generate a complete text description. Furthermore, we calculate the CLIP score between the generated text description and the images produced by each method. As shown in the Table 2, compared to the rendering outputs from graphics engines and images generated by Multi-ControlNet. This demonstrates that our method is capable of generating images that more closely align with the semantic label of 'photo-realism.'

**User study.** We conducted a user study to evaluate the realism of the images produced by various methods. For each graphics engine, we collect 3 scenes, render 8 images from different viewpoints for each scene, and employ both Intrinsic-ControlNet and Multi-ControlNet to generate corresponding images, resulting in a total of 72 image pairs. For each paired image, we ask the participants to rate which image is the most realistic. We collect a total of 28 valid questionnaires and obtain the results shown in Table 2. The table shows the percentage of participants who found the images generated by each method to be the most realistic. The results indicate that images generated by our method are more likely to be perceived as real photographs rather than engine-rendered images. In addition, Multi-ControlNet, due to significant distortions like color shifts, is perceived as having lower realism.

### A.4 MORE COMPARSION

**Comparison with Multi-ControlNet.** We compare our method with the Multi-ControlNet approach. To ensure a fair comparison, we train a separate ControlNet Zhang et al. (2023) for each intrinsic image type, namely albedo, normal, depth, metallicity, roughness, and semantic segmentation, on our entire mixed dataset. Then, We separately use our model and the Multi-ControlNet approach to generate images with various combinations of condition inputs. The comparison results are shown in Figure 12.

**Comparison with graphics engines.** As mentioned in Section 4.2 and Section A.3, we compared our method to several popular graphics engines using the CLIP score. Since the calculation of the CLIP score depends on the provided text prompt, we test a wider variety of prompts here to provide more comprehensive quantitative comparison results. As shown in the Table 5, when we used 'engine rendering style' as the text prompt, the results generated by various graphics engines achieved higher CLIP scores. Conversely, when we used 'photorealism' as the text prompt, our results achieved better scores.

### A.5 TRAINING DETAIL

Our model is trained using 4 NVIDIA A6000 GPUs with a batch size of 24 for 280 iterations based on Stable Diffusion v2.1 pre-trained model (Rombach et al., 2022). For the inference process, we

| Prompt | Building(UE5) | | | Kitchen&Park(Falcor) | | | Alley(blender) | | |
| | a ↑ | a+b ↑ | a+c ↓ | a ↑ | a+b ↑ | a+c ↓ | a ↑ | a+b ↑ | a+c ↓ |
|---|---|---|---|---|---|---|---|---|---|
| Reference | 0.2629 | 0.2529 | 0.2798 | 0.2651 | 0.2630 | 0.2850 | 0.2508 | 0.2801 | 0.2747 |
| Ours | **0.2773** | **0.2703** | **0.2717** | **0.2836** | **0.2822** | **0.2779** | **0.2997** | **0.3036** | **0.2611** |

Table 5: Comparison of the results of our method and different graphic engines.
a: BLIP prompt; b: "photorealism"; c:"engine rendering style".

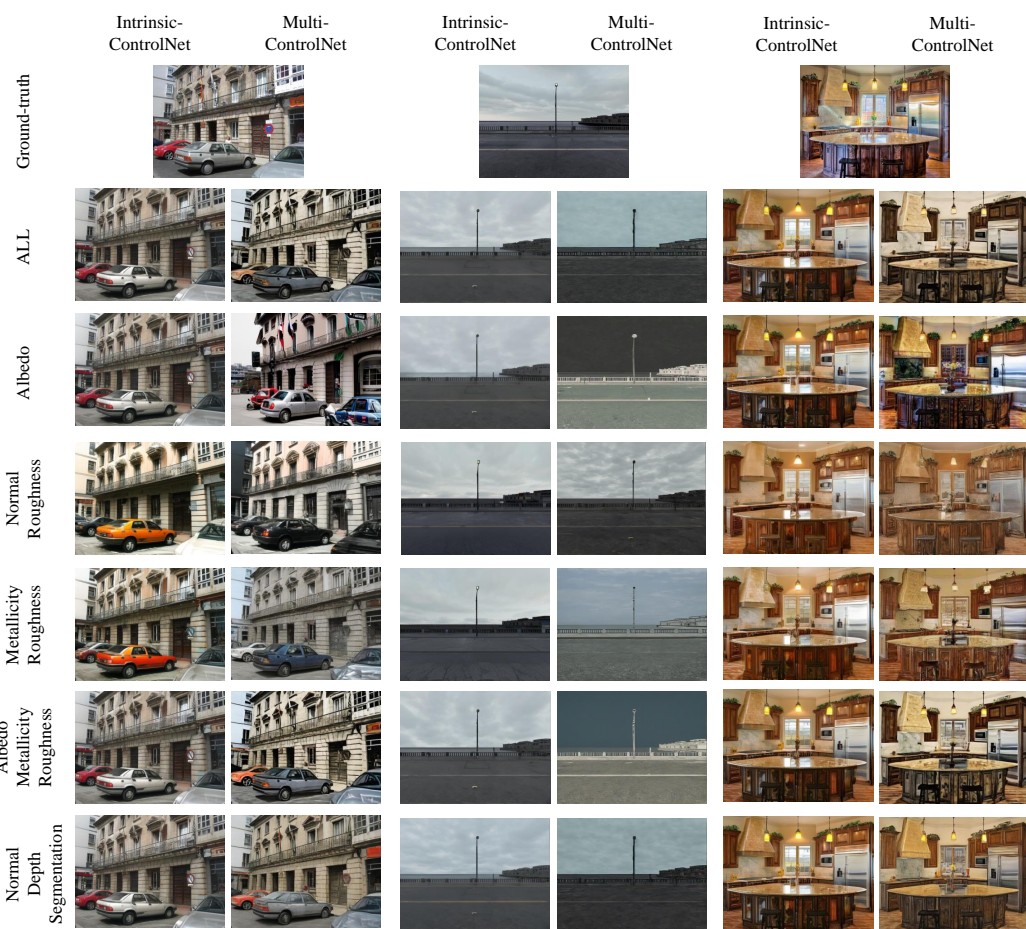

Figure 12: Results of different combination of intrinsic conditions between Intrinsic-ControlNet and Multi-ControlNet.

adopt the DDIM sampler (Song et al., 2020) with 50 sampling steps by using a single NVIDIA A6000 GPU. We use the AdamW optimizer with a fixed learning rate of $1e^{-5}$ (Loshchilov, 2017) and weight decay of 0.01. During training, we center-croped the image with $512 \times 512$ resolution.

## A.6 CONVERGENCE SPEED OF STRUCTURE AND APPEARANCE

Figure 13 shows the intermediate results during the training process for both ControlNet Zhang et al. (2023) and our model. We attempt to use a single branch to control both the structure and appearance of the generated images using ControlNet. During the training process of ControlNet, we can observe that the convergence speeds of appearance and structure differ. As shown in Figure 13, the structure tends to converge much faster than appearance. At an early training stage, the structural features of the generated results are already globally well-controlled, but the color and style are still not precisely managed, with noticeable color errors in the details. As the training

progresses, the structure continues to refine locally, but the slow convergence of appearance causes the color and style to lag behind the ground truth. Since structural control has been achieved, the network as a whole tends to stabilize. This results in the final generated images having issues like being too dark or too magenta. To avoid this issue, we separated the appearance and structure features by using two ControlNet branches with non-shared parameters as mentioned in Section 3.3, which prevents interference between the optimization of appearance and structure.

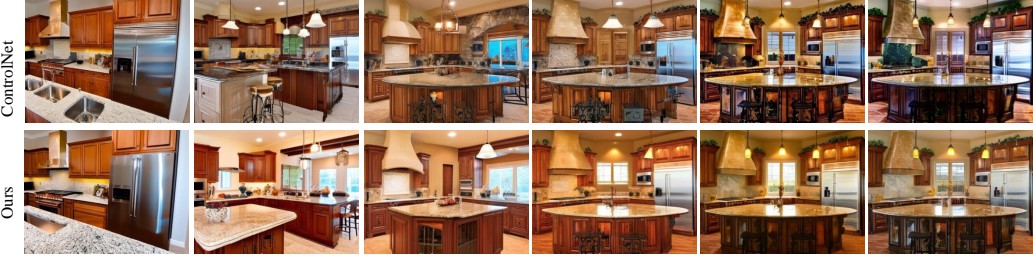

Figure 13: Comparison of the convergence between ControlNet Zhang et al. (2023) and our framework, where ControlNet uses a single branch to control both appearance and structure generation.

### A.7 LIGHT CONTROL WITH PROMPT

In Figure 14, we demonstrate the results of controlling relighting using text prompts. By adding keywords such as weather and time, we can alter the lighting in the generated images. The results not only maintain the structural and appearance features from before the change but also exhibit new lighting effects that are highly realistic and consistent with the real world.

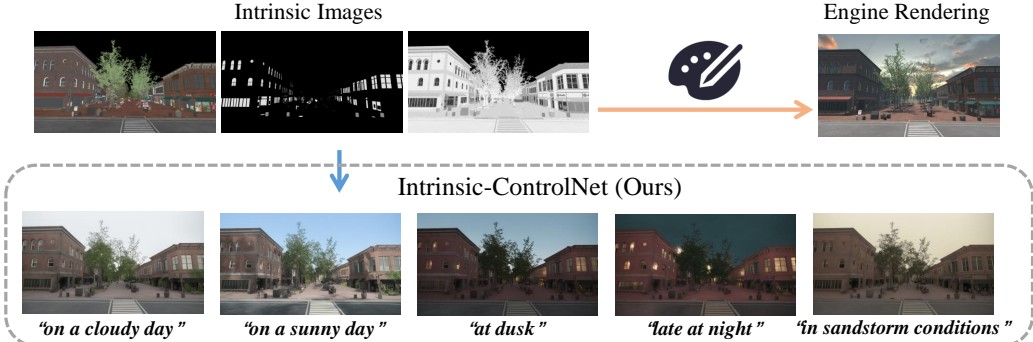

Figure 14: The relighting results of our method. By modifying the condition text prompt, our method can quickly adjust the lighting of the scene to generate realistic relit images.

### A.8 VISUALIZATION OF ENTIRE INTRINSIC IMAGES

We show the entire intrinsic image input achieved from real-world photos in Figure 15, and compare the generated images of our methods with the original photos in Figure 4.

### A.9 MORE DETAILS OF NETWORK ARCHITECTURE

We show the details of network architecture in the Figure 16.

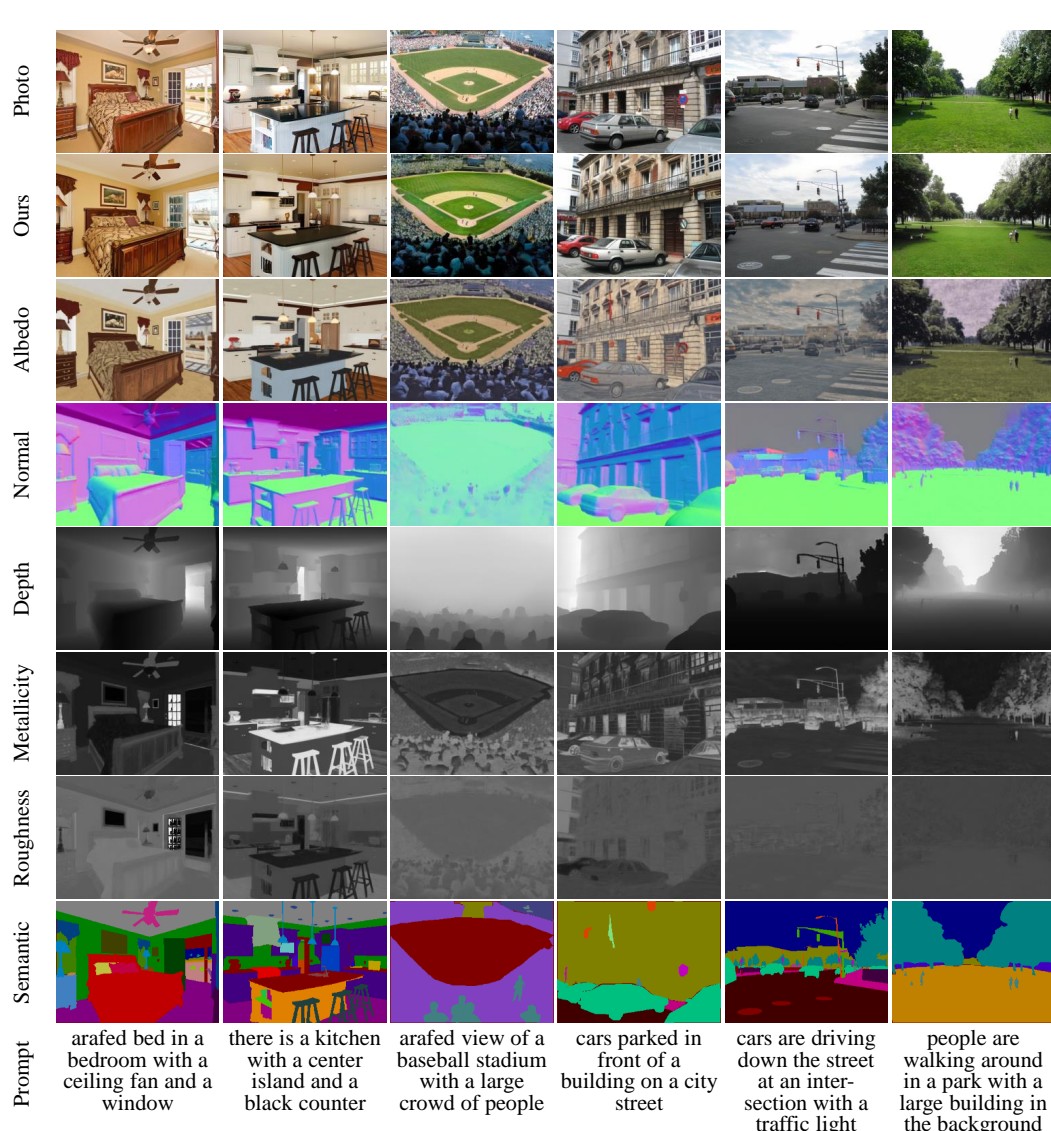

Figure 15: Our framework uses predicted multiple intrinsic images to generate realistic images that resemble the original. Here we show all intrinsic images with the results.

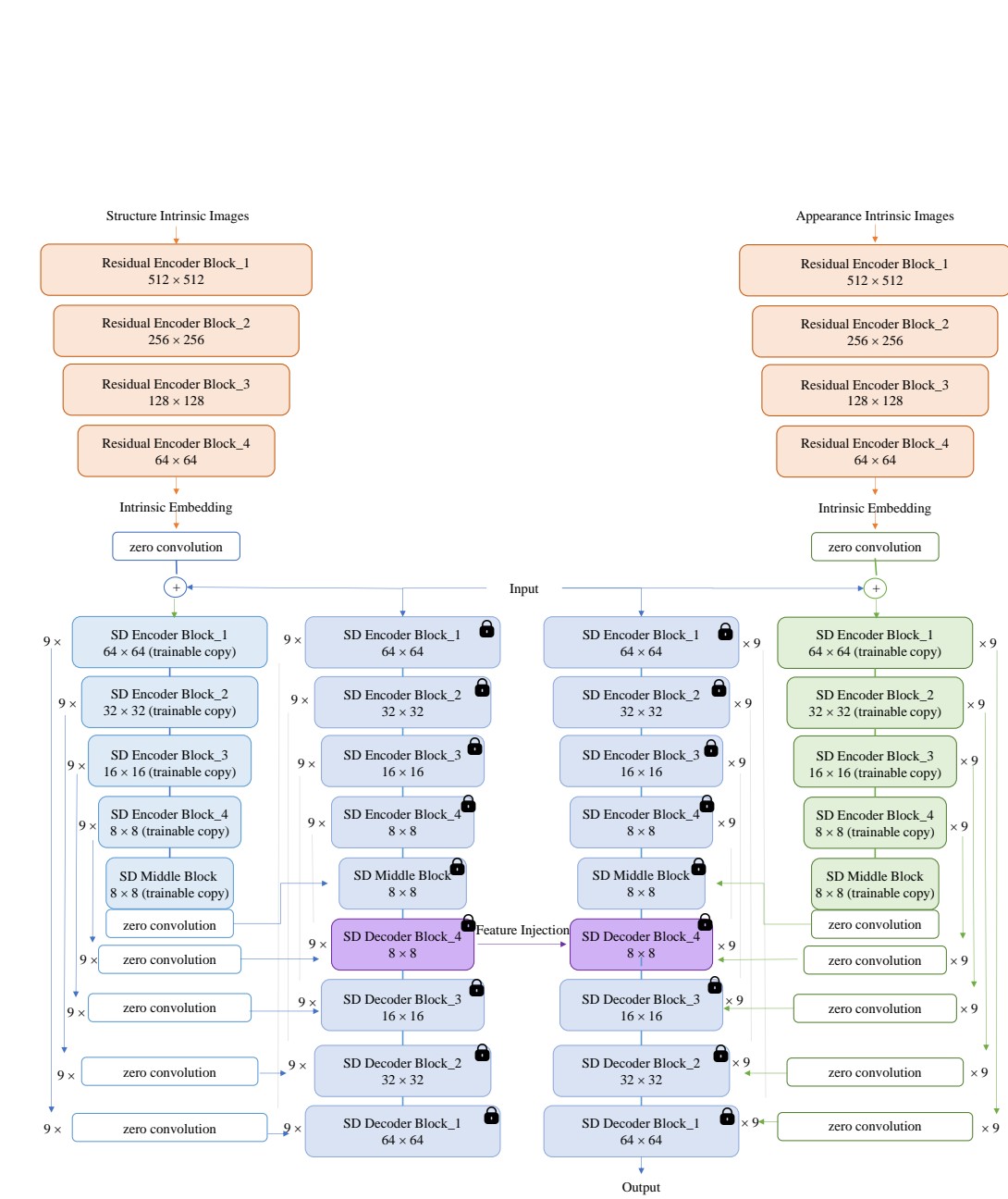

Figure 16: Details of our network architecture.

