# OpenReview forum: "Intrinsic-ControlNet : A Generative Rendering Approach to Render Any Real and Unreal"
_ICLR.cc/2025/Conference — ICLR 2025 Conference Withdrawn Submission_

### Official Review · Reviewer_ZAXX · 2024-10-22

**Soundness:** 3
**Presentation:** 3
**Contribution:** 3
**Rating:** 3
**Confidence:** 5

**Summary:**

This paper focuses on sim-to-real generation, which aims to generate photorealistic images based on normal, albedo, and so on. To achieve this, the paper adapts ControlNet to take multiple condition inputs. For training, the paper proposed to use mixture of both real and synthetic data. This paper gives extensive experiments to show the results and applications.

**Strengths:**

- The paper proposed a novel multi-conditional control method for  any
number of images as input conditions.
- The idea is straight and the writing is easy to follow.

**Weaknesses:**

- Evaluation. Lack of important baselines. This work focuses on the area of sim-to-real generation, but lacks discussion and comparison with many previous works. Specifically, one can render a synthetic image based on the intrinsic conditions with rendering engine, and then perform sim-to-real generation to translate the image to a realistic one. There are many works in this line such as [1]. I suggest the authors to compare with these methods and discuss the pros and cons.

[1] Enhancing Photorealism Enhancement. TPAMI. 2023.

- Speed. When we are talking about the rendering, one main concern is the rendering speed. Rendering engine supports real-time rendering. But the generative method is somehow slow.

- Interaction. Could you show some relighting results? Given the intrinsic, we often need to render the images under different illuminations. This also requires the model to correctly process occlusion and environmental interactions.

- Metrics. Using CLIP score for evaluation is not enough. The authors can consider FID, and aesthetic scores for better evaluation.

- Diversity. The paper shows results of simple scenes. How about complex scenes? Would the details make sense?

- According to ablation results in table 3, the proposed modules seem trivial. The differences in clip scores are only 0.01. More analysis are needed to validate the effectiveness of proposed designs.

**Questions:**

- How do you get albedo in Fig. 7? using pretrained network? What if the predicted albedo is wrong?

---

### Official Review · Reviewer_B98B · 2024-10-29

**Soundness:** 2
**Presentation:** 3
**Contribution:** 3
**Rating:** 5
**Confidence:** 5

**Summary:**

The paper presents a method for conditional image generation based on image intrinsic as input. Specifically, the model leverages a two-pathway ControlNet to condition the pretrained latent diffusion model on normal, depth, segmentation, albedo, metallic, and roughness, focusing on generating photorealistic images given text prompts. The primary technical focus of the paper is to bridge the sim2real gap by training with a mixture of synthetic and real data with designated network pathways. Evaluations are done on both synthetic data and real data using general metrics like CLIP score and user study.

**Strengths:**

- The paper proposes a promising method to bridge the sim2real gap when adding controllability to the pretrained text2image diffusion model. Given the fact that real data has no domain gap but an inaccurate condition-output pair and synthetic data has an accurate condition-output pair but also has a domain gap, the high-level insight of separating different pathways for isolation is inspiring and could be applied to a broader context of adding conditions to the pretrained diffusion model.
- Given intrinsic images of a general scene, the proposed method shows some decent results as a “rendering engine” that renders those inputs into realistic images, although the lighting controllability and 3D consistency are not guaranteed from its current results.
- Ablation studies on different control signals and ControlNet paradigms are carried out to better motivate the proposed method’s designs.
- The paper is well-structured and easy to follow.

**Weaknesses:**

- A bit bold claim: As stated in Section 3.1 (Problem Statement) as well as the introduction, the authors define their framework as generative renderer (Eq. 1). However, I found it to be a bit overclaim for the following reasons: 1) There is no qualitative result to show the 3D consistency / temporal consistency of the generated results; and 2) As specified in Eq. 1, the lighting condition seems to be the input to the model, however, I do not find any lighting condition fed into the model throughout the method design. Therefore, generative image stylization could be a better and more accurate terminology for the proposed method.
- Strong assumption: the proposed method assumes the known intrinsic of images. However, in practice, those intrinsic images are predicted from the pretrained model (Intrinsic Diffusion) which could lead to unexpected errors for an arbitrary in-the-wild image. Given the fact that the intrinsic decomposition of in-the-wild images is still a challenging task that is unsolved, I would not prefer a model that relies on accurate image intrinsic as input.
- Limited quality: Although the proposed method shows overall intrinsic-conditioned stylization results in Fig. 4, 6, 7, I do observe severe artifacts when zooming in. For example, the car wheel is distorted and the appearance of buildings/traffic lights is also distorted. This could be better illustrated if the authors could provide full-resolution results. As a “generative renderer”, people do not want the renderer to alter the original content they fed in.

**Questions:**

Besides the weaknesses mentioned above, I have the following questions regarding details:

- The motivation of multi-way condition switch instead of direct concatenation? I feel like a direct concatenation and conditional encoder can also fuse multiple conditions into a unified one without specific weighting designs.
- Where is the lighting condition in the model as specified in Eq. 1?
- How do you determine the weights of each modality’s output for the Multi-ControlNet baseline?

---

### Official Review · Reviewer_uweV · 2024-11-02

**Soundness:** 3
**Presentation:** 3
**Contribution:** 2
**Rating:** 5
**Confidence:** 4

**Summary:**

This paper introduces Intrinsic-ControlNet, a rendering engine-like generative framework that generates realistic RGB images by taking intrinsic images as network inputs. In particular, the authors propose a novel multi-conditional control method, allowing the model to accept any combination of intrinsic images as input. Moreover, to ensure the appearance of realism, the training process only incorporates appearance conditions derived from real-world images. Experimental results demonstrate the efficacy of Intrinsic-ControlNet in generating controllable and photorealistic images that are conditioned on the input intrinsic images.

**Strengths:**

1. This paper proposes a valid generative rendering framework for the controllable and editable generation of photorealistic RGB images based on the input intrinsic images.
2. The novel multi-conditional control method allows for flexibility in input conditions.
3. The separation of appearance and structural conditions into different ControlNets ensures the realistic appearance of generated images.
4. This innovative framework supports various applications like flexible scene editing and object insertion.

**Weaknesses:**

1. A notable limitation of the Intrinsic-ControlNet approach is the inevitable domain gap that arises from the training data. Specifically, the network is only trained on appearance condition images predicted from real images using pretrained diffusion models, whereas in practical applications, it is expected to generate images conditioned on appearance intrinsic images from synthetic images. As discussed in lines 283-285, this domain gap can lead to degenerate results.
2. The generated images exhibit unauthentic characteristics. For instance, the lighting of inserted objects in Figure 7 appears unnatural. Furthermore, the second column of Figure 8 reveals misalignment between the generated image and the normal after editing. Additionally, the third column shows Intrinsic-ControlNet hallucinating an unrealistic light on the floor, which does not correspond to the ceiling lights. Moreover, despite the authors' claim that their method can eliminate environmental lighting effects, the resulting image in the fourth column still retains indirect lighting from the two lamps.
3. Unlike traditional rendering engines, the generation process of Intrinsic-ControlNet inherently involves a significant degree of randomness. Consequently, ensuring 3D consistency in the generated images becomes challenging, particularly when rendering 3D assets from a continuous rendering trajectory. This can result in a jittering effect in the rendering output.

**Questions:**

The authors of the abstract lament the high computational cost and expense associated with traditional simulations. However, it is surprising that they fail to provide a comparative analysis of the efficiency and cost-effectiveness of their proposed model versus traditional simulations in the subsequent paper.

---

### Official Review · Reviewer_W6i7 · 2024-11-02

**Soundness:** 3
**Presentation:** 2
**Contribution:** 2
**Rating:** 5
**Confidence:** 4

**Summary:**

This paper proposes a novel approach for rendering photo-realistic images via a generative framework instead of traditional physical-based rendering methods. The method employs a multi-conditioned ControlNet that takes varying number of intrinsic images as input and produces rendered images. This method is trained and evaluated on both synthetic and real-world datasets. Ablation study is conducted to examine design choices (e.g. separate structure and appearance condition, architecture of encoder ).

**Strengths:**

1. Proposes an alternative approach to photorealistic image synthesis that bypasses the traditional graphics pipeline.

2. Effectively leverages real-world data, reducing the bias towards synthetic data.

3. Comprehensive ablation studies

**Weaknesses:**

1. Lack of Comparative Analysis with RGB↔X. The paper does not compare the proposed method to RGB↔X or a re-implemented version of it despite the similarity in task setting and both methods use generative framework. Additionally, the authors mention that "existing methods (e.g., RGB↔X) face challenges with multiple control conditions as inputs," but these challenges are not fully explained. This absence leaves the effectiveness of the multi-conditioned ControlNet uncertain without direct comparison to RGB↔X.

2. Questionable Evaluation Metrics and Comparisons

    1. Effectiveness of CLIP Metrics. In Table 1, this method outperforms ground truth (GT) in CLIP metrics, raising doubts about whether CLIP can reliably assess the quality of rendered images. CLIP primarily measures text-image alignment, which may not adequately capture "accuracy" or quality in rendered images.

    2. Comparison with Render Engines. The comparison with traditional render engines lacks clarity, especially regarding the environment maps used. One text prompt (e.g. "a sunny day") could have different and varied environment maps which play a crucial role in final rendered image. In the last column of Fig. 6, the render engine appears to use a high-contrast environment map, resulting in a "less realistic" outcome. Additionally, the first two columns in Fig. 6 show sunset clouds in render engine results, which influence CLIP similarity scores but are not included in the text prompt, potentially skewing comparisons.

**Questions:**

1. The study only ablates three combinations of intrinsic images (A, M+R, A+N+D+S), leaving some questions about the individual contributions of each intrinsic type. For instance, how does depth information contribute when normal information is already provided, as depth can often be inferred from normals? Similarly, how does the semantic map influence the final result, given that traditional graphics engines do not typically require a semantic map for rendering? Further ablation would be beneficial.

**Details Of Ethics Concerns:**

No ethical concerns have been identified.

---

### Note · Authors · 2024-11-14

I have read and agree with the venue's withdrawal policy on behalf of myself and my co-authors.